# BPQP: A Differentiable Convex Optimization Framework for Efficient End-to-End Learning

## Abstract

Real-world decision-making processes often employ a two-stage approach, where a machine learning model first predicts key parameters, followed by a constrained convex optimization model to render final decisions. The machine learning model is typically trained separately to minimize prediction error, which may not necessarily align with the ultimate goal, resulting in potentially suboptimal decisions. The predict-then-optimize approach offers an end-to-end learning solution to bridge this gap, wherein machine learning models are trained in tandem with the optimization model to minimize the ultimate decision error. However, practical applications involving large-scale datasets bring about significant challenges due to the inherent need for efficiency to fully realize the potential of the predict-then-optimize approach. Although recent works have started to focus on predict-then-optimize, they have been limited to small-scale datasets due to low efficiency. In this paper, we propose BPQP, a differentiable convex optimization framework for efficient end-to-end learning. To address the challenge of efficiency, we initially reformulate the backward pass as a simplified and decoupled quadratic programming problem by exploiting the structural trait of the KKT matrix, followed by solving it using first-order optimization algorithms. Extensive experiments on both simulated and real-world datasets have been conducted, demonstrating a considerable improvement in terms of efficiency – at least an order of magnitude faster in overall execution time. We significantly improve efficiency and highlight the superiority of BPQP compared to baselines, including the traditional two-stage learning approach.

## 1 Introduction

Data-driven stochastic optimization often relies on a two-stage solution: first, it reduces uncertainty by predicting key unknown parameters based on available contextual features, then it utilizes these predictions for downstream constrained optimization. The *predict-then-optimize* paradigm Elmachtoub & Grigas (2022); Wilder et al. (2019); Liu & Grigas (2021) integrates these two stages, enabling end-to-end training to directly minimize *regret* – the difference between the decision made from the prediction and the optimal decision in hindsight Kotary et al. (2021); Mandi et al. (2020). This paradigm, and closely related data-driven optimization methods Agrawal et al. (2019b;a); Amos & Kolter (2017), have proven effective in various applications. Here, we focus on convex optimization because of its wide applications in portfolio optimizationWilder et al. (2019), control systemsGuo & Wang (2010), signal processingMattingley & Boyd (2010), and more.

Training such an end-to-end model necessitates the incorporation of external differentiable convex optimization layers into the training loop of a machine learning (ML) model. Optimization problems typically do not have a general closed-form solution and require more sophisticated solutions. These solutions can be categorized into explicit and implicit methods based on whether an explicit computational graph is constructed. Explicit methods Domke (2012); Blondel et al. (2021); Foo et al. (2007); Sun et al. (2022) unroll the iterations of the optimization process, incurring additional costs. On the other hand, implicit methods utilize the Implicit Function Theorem to derive the gradients. Some of them Amos & Kolter (2017); Agrawal et al. (2019a;b) are designed for specific problems, which restrict the options for forward optimization and deteriorates efficiency. On the other hand,

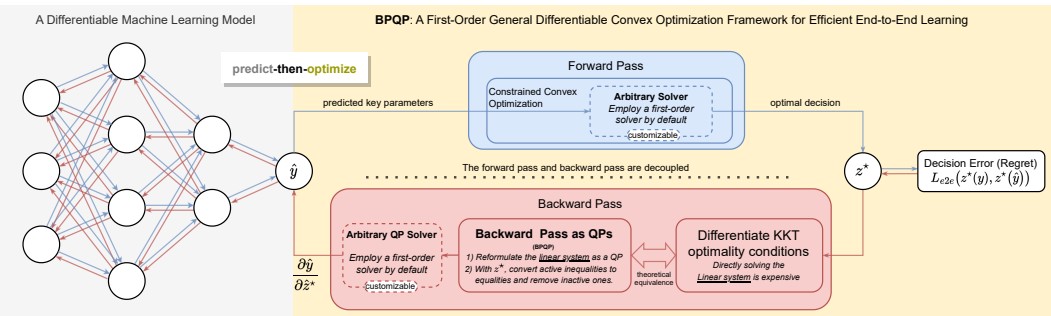

Figure 1: The learning process of BPQP: the machine learning model outputs key parameters $\hat{y}$ and then generates the optimal decision $z^\star$ in the forward pass; the backward pass propagates the decision error to the machine learning model for end-to-end learning; the process is accelerated by reformulating and simplifying the problem first and then adopting efficient solvers.

other approaches Gould et al. (2021); Blondel et al. (2021) propose more general solutions but are not efficient in the backward pass. There is still plenty of room for improvement in terms of efficiency. To enable rapid, tractable differentiable convex optimization layer and further expand the capabilities of the predict-then-optimize paradigm, we propose a general, first-order differentiable convex framework for large-scale end-to-end learning, namely BPQP.

Specifically, we simplify the backward pass by reformulating it into a simpler QP problem, which we refer to as the **B**ackward **P**ass as a **Q**uadratic **P**rogramming (BPQP). This decouples the forward and backward passes and creates a framework that can leverage existing efficient solvers (with the first-order solver, Alternating Direction Method of Multipliers (ADMM) Stellato et al. (2020), as the default) that do not require differentiability in both passes. Simplifying and decoupling the backward pass significantly reduces the computational cost in both the forward and backward passes. This key idea is summarized in Fig. 1.

Our proposed framework has several theoretical and practical contributions:

**Efficient Gradients Computation:** Empirically, BPQP significantly improves the overall computational time, achieving up to $21.17\times$, $16.17\times$, and $1.67\times$ faster performance over existing differentiable layers on 100-dimension Linear Programming, Quadratic Programming, and Second Order Cone Programming, respectively. Furthermore, when applied to large-scale real-world portfolio optimization, BPQP enhances the Sharpe ratio from 0.65(±0.25) to 1.28(±0.43) compared to widely-adopted methods designed for the two-stage approach.

**Flexible Solver Choice:** BPQP accommodates any general-purpose convex solver to integrate the differentiable layer for end-to-end training. In addition, we propose a specialized method for the backward pass: Backward Pass as Quadratic Programming (BPQP). This method leverages structural traits such as sparsity, solution polishing Stellato et al. (2020), and active-sets Wolfe (1959) for efficient and accurate gradients computation. The method uses Quadratic Programming (QP) to avoid the inversion of the KKT matrix and enables *large-scale gradients computation* via the Alternating Direction Method of Multipliers (ADMM). This flexibility in solver choice allows for better matching of solver capabilities with specific problem structures, potentially leading to improved efficiency and performance.

## 2 RELATED WORKS

**Explicit methods** Optimization problems typically do not have a general closed-form solution formula that expresses the decision variable in terms of other parameters. To address this challenge, explicit methods Domke (2012); Blondel et al. (2021); Foo et al. (2007) unroll the iterations of the optimization process and use the decision variable from the final iteration as a proxy solution for the optimization problem. This constructs an explicit computational graph from the parameters to the proxy parameters. Typically, these methods are designed for unconstrained optimizations. Applying them directly to constrained optimizations is computationally expensive because it requires project-

ing decision variables into a feasible region. Alt-Diff Sun et al. (2022) is a novel unrolling solution that decouples constraints from the optimization and significantly reduces the computational cost. While advanced unrolling methods continue to improve their efficiency, they require an additional cost in the unrolled computational graph that increases with the number of optimization iterations.

**Implicit methods** In contrast, implicit methods use the Implicit Function Theorem to relate the decision variable to other parameters. These methods specifically apply the theorem to KKT conditions in convex optimization. Some works are designed for specific problems, which limits the choices of forward optimization and deteriorates efficiency. OptNet Amos & Kolter (2017) presented a differentiable batched-GPU QP solver. diffcp Agrawal et al. (2019a;b) considers computing the derivative of a convex cone program by implicitly differentiating the residual map for its homogeneous self-dual embedding. Open-source convex solver CVXPY Diamond & Boyd (2016) adopts a similar method and computes gradients by SCS O'donoghue et al. (2016). Another line of work uses more general solutions, which are not efficient enough to handle the backward pass. Gould et al. (2021) decouples the forward and backward pass. JaxOpt Blondel et al. (2021) proposes a simple approach to adding implicit differentiation on top of any existing solver, which significantly lowers the barrier to using implicit differentiation. Our work, BPQP, is based on implicit methods. First, we simplify the backward pass by reformulating it into a simpler decoupled QP problem. Problem simplification and decoupling greatly reduce the computational cost in both the forward and backward passes.

**Learning-to-optimize** Existing work on *Learn-to-Optimize* trains an approximated solver network (e.g., Donti et al. (2021); Cristian et al. (2023); Kong et al. (2022)) . This approach provides solutions as efficient as closed-form solutions. However, these methods either have low accuracy or only perform well in specific scenarios, which is outside the scope of our research. Therefore, the Appendix A.6 includes additional discussions about approximate and scenario-specific methods.

## 3 BACKGROUND

### 3.1 PREDICT-THEN-OPTIMIZE FRAMEWORK

In this section, we formally describe the predict-then-optimize framework for stochastic decision making problems. We assume that the problem of our interest has convex objective and constraints, but the key parameter $y \in \mathbb{R}^p$ is not observable when the decision is made. For each optimization instance, a prediction of $y$ is required to solve the downstream deterministic optimization problem. Specifically, let $(x \in \mathcal{X}, y \in \mathcal{Y}) \sim \mathcal{D}$ denote standard input-output pairs drawn from the real and unknown distribution $\mathcal{D}$. Suppose a ML model $\mathcal{N}$, parameterized by $\theta$, with input features $x$ is trained to generate such a prediction $\hat{y} = \mathcal{N}(x; \theta) = \mathbb{E}_{y \sim p_\theta(y|x)}[y]$, namely $\hat{y} \in \mathbb{R}^p = \mathbb{E}[y|x]$. Let $z_{\hat{y}} \in \mathbb{R}^d$ denote the decision variable of the corresponding optimization relying on random parameter $\hat{y}$. The parameterized convex optimization can be formalized as follows:

$$z_{\hat{y}}^\star = \arg\min_{z \in \mathbb{R}^d} f_{\hat{y}}(z) \quad \text{subject to } h_{\hat{y}}(z) = 0, \ g_{\hat{y}}(z) \leq 0, \tag{1}$$

For any given $\hat{y}$, $f_{\hat{y}}(\cdot) : \mathbb{R}^d \to \mathbb{R}$ the $C^2$ continuous convex objective function, and $h_{\hat{y}}(\cdot) : \mathbb{R}^d \to \mathbb{R}^n$, $g_{\hat{y}}(\cdot) : \mathbb{R}^d \to \mathbb{R}^m$ the $n$-dimension equality constraints and $m$-dimension inequality constraints representing the feasible region. $h$ and $g$ are both $C^2$ continuous convex functions. As we demonstrated, the optimal decision $z^\star$ is a random variable depending on $\hat{y}$, $z_{\hat{y} \sim p_\theta(y|x)}^\star$.

To implement an end-to-end approach training for $\mathcal{N}$, upon observing the optimal decision $z_y^\star$ relative to the true instantiation of $x$ and $y$, we update the parameterized model $\mathcal{N}(x; \theta)$ correspondingly, minimizing regret. The overall end-to-end training procedure can be viewed as maximizing posterior probability given decision error and prediction error.

$$p(\theta|regret, y, x) \propto \underbrace{p(regret|y, x, \theta)}_{\text{Decision Error}} \underbrace{p(y|x, \theta)p(x|\theta)}_{\text{Prediction Error}} \underbrace{p(\theta)}_{\text{prior}}, \tag{2}$$

Ideally, our goal here is to use supervised learning to predict the unspecified parameter $\hat{y}$ from empirical data in ways that the decisions made from estimation $z_{\hat{y}}^\star$ match the best decisions taken in hindsight $z_y^\star$, i.e., regret

$$\text{regret}(y, \hat{y}) = f_y\left(z_{\hat{y}}^\star\right) - f_y\left(z_y^\star\right), \tag{3}$$

Given the realized parameter $y$, Chen et al. (2022) found the exact optimization decision error empirically to be a narrow (Dirac-like) target distribution centered at the ground truth $regret = 0$. The rest of the terms above can be viewed as *prior* distribution forms the classic prediction error of which we choose simple MSE loss, yielding the simplified end-to-end (predict-then-optimize) loss, weighted by constant $\beta \in (0, 1)$:

$$\mathcal{L}_{e2e} = \beta \underbrace{\mathbb{E}_{x,y\sim\mathcal{D}} \left[ \left\| f_y\left(z_{\hat{y}}^{\star}\right) - f_y\left(z_y^{\star}\right) \right\|^2 \right]}_{\text{Decision Error: } regret} + \underbrace{\mathbb{E}_{x,y\sim\mathcal{D}} \left[ \|y - \hat{y}\|^2 \right]}_{\text{Prediction Error}} + \alpha \underbrace{\mathcal{L}_{reg}(\theta)}_{\text{prior}}. \tag{4}$$

**Comparison to Two-stage Approach** The two terms in Eq. (4) are concerned with decision error and prediction error. The former is often approximated as surrogate loss due to the complexity of computing regret in previous work Elmachtoub & Grigas (2022); Wilder et al. (2019). But surrogate loss is sub-optimal and often cannot handle learning feasible solutions of complex constraints over thousands or even hundreds of dimensions. Relatively, the traditional Two-stage approach divides stochastic optimization into two separate stages: first train a prediction model on $y$ and then solve the optimization problems $z_{\mathbb{E}[y|x]}^{\star}$ separately. The shortcoming of the Two-stage approach is that it does not take the effect on the optimization task into account. Training to minimize Two-stage loss (prediction loss) is not guaranteed to deliver better performance in terms of the decision problem Mandi et al. (2020); Ifrim et al. (2012). As a special case of end-to-end loss, we conclude that the Two-stage approach minimizes a lower bound of the total end-to-end loss and does not necessarily result in the minimization of regret.

$$\mathcal{L}_{2stage} = \mathbb{E}_{x,y\sim\mathcal{D}}[\|y - \hat{y}\|^2] \leq \mathcal{L}_{e2e}. \tag{5}$$

## 3.2 DIFFERENTIATING THROUGH KKT CONDITIONS

One major challenge of adopting the predict-then-optimize approach is to backpropagate losses through the argmin operator, namely the backward pass.

$$\frac{\partial\mathcal{L}}{\partial y} = \frac{\partial\mathcal{L}}{\partial z^{\star}} \frac{\partial z^{\star}}{\partial y}, \tag{6}$$

We consider a general convex problem in Eq. (1). To compute the derivative of the solution $z^{\star}$ to parameter $y$, OptNet Amos & Kolter (2017) differentiates the KKT conditions using techniques from matrix differential calculus. Following this method, the Lagrangian is given by (omitting $y$),

$$L(z, \nu, \lambda) = f(z) + \nu^{\top} h(z) + \lambda^{\top} g(z), \tag{7}$$

where $\nu \in \mathbb{R}^m$ and $\lambda \in \mathbb{R}^n$, $\lambda \geq 0$ respectively denotes the dual variables on the equality and inequality constraints. The sufficient and necessary conditions for optimality of Eq. (1) are KKT conditions. Applying the Implicit Function Theorem (IFT) to the KKT conditions and let $P(z^{\star}, \nu^{\star}, \lambda^{\star}) = \nabla^2 f(z^{\star}) + \nabla^2 h(z^{\star})\nu^{\star} + \nabla^2 g(z^{\star})\lambda^{\star}$, $A(z^{\star}) = \nabla h(z^{\star})$ and $G(z^{\star}) = \nabla g(z^{\star})$. Let $q(z^{\star}, \nu^{\star}, \lambda^{\star}) = \partial(\nabla f(z^{\star}) + \nabla h(z^{\star})\nu^{\star} + \nabla g(z^{\star})\lambda^{\star})/\partial y$, $b(z^{\star}) = \partial h(z^{\star})/\partial y$ and $c(z^{\star}, \lambda^{\star}) = \partial(D(\lambda^{\star})g(z^{\star}))/\partial y$. Then the matrix form of the linear system can be written as:

$$\begin{bmatrix} P(z^{\star}, \nu^{\star}, \lambda^{\star}) & G(z^{\star})^{\top} & A(z^{\star})^{\top} \\ D(\lambda^{\star}) G(z^{\star}) & D(g(x^{\star})) & 0 \\ A(z^{\star}) & 0 & 0 \end{bmatrix} \begin{bmatrix} \frac{\partial z^{\star}}{\partial y} \\ \frac{\partial \lambda^{\star}}{\partial y} \\ \frac{\partial \nu^{\star}}{\partial y} \end{bmatrix} = - \begin{bmatrix} q(z^{\star}, \nu^{\star}, \lambda^{\star}) \\ c(z^{\star}, \lambda^{\star}) \\ b(z^{\star}) \end{bmatrix}, \tag{8}$$

$D(\cdot) : \mathbb{R}^m \to \mathbb{R}^{m\times m}$ represents a diagonal matrix that formed from a vector and $z^{\star}, \nu^{\star}, \lambda^{\star}$ denotes the optimal primal and dual variables. Left-hand side is the KKT matrix of the original optimization problem times the Jacobian matrix of primal and dual variables to the omitted parameter $y$, e.g., $\frac{\partial z^{\star}}{\partial y} \in \mathbb{R}^{p\times d}$. Right-hand side is the negative partial derivatives of KKT conditions to the $y$.

We can then backpropagate losses by solving the linear system in Eq. (8). In practice, however, explicitly computing the actual Jacobian matrices $\frac{\partial z^{\star}}{\partial y}$ is not desirable due to space complexity; instead, Amos & Kolter (2017) products previous pass gradient vectors $\frac{\partial\mathcal{L}}{\partial z^{\star}} \in \mathbb{R}^d$, to reform it by notations $[\tilde{z} \in \mathbb{R}^d, \tilde{\lambda} \in \mathbb{R}^m, \tilde{\nu} \in \mathbb{R}^n]$ (see Appendix A.2):

$$\begin{bmatrix} P(z^{\star}, \nu^{\star}, \lambda^{\star}) & G(z^{\star})^{\top} & A(z^{\star})^{\top} \\ D(\lambda^{\star}) G(z^{\star}) & D(g(x^{\star})) & 0 \\ A(z^{\star}) & 0 & 0 \end{bmatrix} \begin{bmatrix} \tilde{z} \\ \tilde{\lambda} \\ \tilde{\nu} \end{bmatrix} = - \begin{bmatrix} (\frac{\partial\mathcal{L}}{\partial z^{\star}})^{\top} \\ 0 \\ 0 \end{bmatrix}. \tag{9}$$

And the direct gradients $\nabla_y \mathcal{L} \in \mathbb{R}^p = [q(z^{\star}, \nu^{\star}, \lambda^{\star}), c(z^{\star}, \lambda^{\star}), b(z^{\star})][\tilde{z}, \tilde{\lambda}, \tilde{\nu}]^{\top}$.

## 4 METHODOLOGY

### 4.1 BACKWARD PASS AS QPS

Our method solves Eq. (9) using reformulation method. Consider a general class of QPs that have $d$ decision variables, $n$ equality constraints and $m$ inequality constraints:

$$\underset{\tilde{z}}{\text{minimize}} \frac{1}{2}\tilde{z}^\top P\tilde{z} + q^\top \tilde{z} \quad s.t.\ A\tilde{z} = b,\ G\tilde{z} \leq c, \tag{10}$$

where $P \in \mathbb{S}_+^d$, $q \in \mathbb{R}^d$, $A \in \mathbb{R}^{n\times d}$, $b \in \mathbb{R}^n$, $G \in \mathbb{R}^{m\times d}$ and $c \in \mathbb{R}^m$. KKT conditions write down in matrix form:

$$\begin{bmatrix} P & G^\top & A^\top \\ D(\tilde{\lambda})G & D(G\tilde{z} - c) & 0 \\ A & 0 & 0 \end{bmatrix} \begin{bmatrix} \tilde{z} \\ \tilde{\lambda} \\ \tilde{\nu} \end{bmatrix} = \begin{bmatrix} -q \\ D(\tilde{\lambda})c \\ b \end{bmatrix}. \tag{11}$$

We note that Eq. (11) is equivalent to Eq. (9) if and only if: (i) $P = P(z^\star, \nu^\star, \lambda^\star)$, $A = A(z^\star)$, $D(\tilde{\lambda})G = D(\lambda^\star)G(z^\star)$, $[-q, D(\tilde{\lambda})c, b] = [-\left(\frac{\partial \mathcal{L}}{\partial z^\star}\right)^\top, 0, 0]$ and (ii) $P(z^\star, \nu^\star, \lambda^\star)$ is positive semi-definite. As the backward pass solves after the forward pass, we can change inequality constraints to an accurate active-set (i.e., a set of binding constraints) of equality conditions, and then condition (i) always holds for the equality-constrained QP. From this, the following theorem can be obtained

**Theorem 1** *Suppose that the convex optimization 1 is not primal infeasible and the corresponding Jacobian vector $\nabla_y \mathcal{L}$ exists. It is given by $\nabla_y \mathcal{L} = [q(z^\star, \nu^\star, \lambda^\star), c(z^\star, \lambda^\star), b(z^\star)][\tilde{z}, \tilde{\lambda}, \tilde{\nu}]^\top$ and $\tilde{z}, \tilde{\lambda}, \tilde{\nu}$ is the optimal solution of following equality constrained Quadratic Problem:*

$$\underset{\tilde{z}}{\text{minimize}} \frac{1}{2}\tilde{z}^\top P\tilde{z} + q^\top \tilde{z} \quad s.t.\ A\tilde{z} = b,\ G_+\tilde{z} = c_+. \tag{12}$$

*Where $P = P(z^\star, \nu^\star, \lambda^\star)$, $A = A(z^\star)$, $G_+ = G_+(z^\star)$ and $[-q, c_+, b] = [-\left(\frac{\partial \mathcal{L}}{\partial z^\star}\right)^\top, 0, 0]$. $G_+, c_+$ has the same row of active-set as original inequality constraints.*

Though our BPQP procedure described above also applies to Jacobians with forms other than vectors, e.g., matrices, in these cases where each 1-dimension column in $[\tilde{z}, \tilde{\lambda}, \tilde{\nu}]^\top$ right multiply the same KKT matrix and can be viewed as QPs packed in multi-dimensions, directly calculating the *inverse* of the KKT matrix may be more appropriate, especially when it contains a special structure like OptNet Amos & Kolter (2017) and SATNet Wang et al. (2019).

**General Gradients** The intuition of BPQP is that the linearity of IFT requires the KKT matrix left-multiply homogeneous linear partial derivative variables. Theorem 1 highlights a special situation that considers gradients at the optimal point (where KKT conditions are satisfied). Generally, BPQP provides perspective to define gradients in parameter-solution space that preserves KKT norm. Let us consider a series of vectors denoting the $k$th iteration norm value of KKT conditions:

$$\|r^{(k)}\| = \| \left( r_{dual}^{(k)}, r_{cent}^{(k)}, r_{prim}^{(k)} \right) \| = C_k. \tag{13}$$

Where $r^{(k)} \in \mathbb{R}^{d+m+n}$ the KKT conditions in $k$th iteration and $C_k \in \mathbb{R}$ the norm value. The series $\{C_0, C_1, ..., C_k\}$ converges to 0 if the iteration algorithm is a contraction operator. Let $\mathcal{Q}^{(k)}$ denote standard QP problem w.r.t. parameter $P_k, q_k, A_k, b_k, G_k, c_k$ and decision variable $z_k$. At each iteration, BPQP yields $\nabla_y \mathcal{L}^{(k)}$ that preserves $\|r^{(k)}\| = C_k$. (See in Appendix A.3)

**Time Complexity** The time complexity of solving such QP is $\mathcal{O}(N^3)$ in the number of variables and constraints which is at the same level as directly solving the linear system Eq. (9). However, reformulation as QP provides substantial structures that can be exploited for efficiency, such that (we cover them in Section 4.2) sparse matrix, solution polishing Stellato et al. (2020), active-sets, and first-order methods, etc. Cleverly implement BPQP, experiments at fairly large-scale dimensions in practice highlight BPQP's capacity in comparison to the state-of-art differentiable solver and NN-based optimization layers. Intuitively, BPQP is more efficient than previous methods because it utilizes the convex QP structural trait in the backward pass.

## 4.2 EFFICIENTLY SOLVE BACKWARD PASS PROBLEM WITH OSQP

The solver we referenced is OSQP Stellato et al. (2020), which incorporates the sparse matrix method and uses a first-order Alternating Direction Method of Multipliers (ADMM) method to solve QPs. We summarize OSQP here Ichnowski et al. (2021). On each iteration, it refines a solution from an initialization point for vectors $z^{(0)} \in \mathbb{R}^d$, $\lambda^{(0)} \in \mathbb{R}^m$, and $\nu^{(0)} \in \mathbb{R}^n$. And then iteratively computes the values for the $k+1$th iterates by solving the following linear system:

$$\begin{bmatrix} P + \sigma I & A^\top \\ A & \mathrm{diag}(\rho)^{-1} \end{bmatrix} \begin{bmatrix} z^{(k+1)} \\ v^{(k+1)} \end{bmatrix} = \begin{bmatrix} \sigma z^{(k)} - q \\ \lambda^{(k)} - \mathrm{diag}(\rho)^{-1} \nu^{(k)} \end{bmatrix}, \tag{14}$$

And then performing the following updates:

$$\begin{aligned} \tilde{\lambda}^{(k+1)} &\leftarrow \lambda^{(k)} + \mathrm{diag}(\rho)^{-1} \left( v^{(k+1)} - \nu^{(k)} \right) \\ \lambda^{(k+1)} &\leftarrow \Pi \left( \tilde{\lambda}^{(k+1)} + \mathrm{diag}(\rho)^{-1} \nu^{(k)} \right) \\ \nu^{(k+1)} &\leftarrow \nu^{(k)} + \mathrm{diag}(\rho) \left( \tilde{\lambda}^{(k+1)} - \lambda^{(k+1)} \right) \end{aligned} \tag{15}$$

where $\sigma \in \mathbb{R}_+$ and $\rho \in \mathbb{R}_+^n$ are the *step-size* parameters, and $\Pi : \mathbb{R}^m \to \mathbb{R}^m$ denotes the Euclidean projection onto constraints set. When the primal and dual residual vectors are small enough in norm after $k$th iterations, $z^{(k+1)}, \lambda^{(k+1)}$ and $\nu^{(k+1)}$ converges to exact solution $z^\star, \lambda^\star$ and $\nu^\star$.

In particular, given a backward pass problem Eq. (12) with known active constraints, as stated in OSQP, we form a KKT matrix below[1]:

$$\begin{bmatrix} P + \delta I & G_+^\top & A^\top \\ G_+ & -\delta I & 0 \\ A & 0 & -\delta I \end{bmatrix} \begin{bmatrix} \tilde{z} \\ \tilde{\lambda}_+ \\ \tilde{\nu} \end{bmatrix} = \begin{bmatrix} -q \\ 0 \\ 0 \end{bmatrix}, \tag{16}$$

As the original KKT matrix is not always inveritible, e.g., if it has one or more redundant constraints, we modify it to be more robust for QPs of all kinds by adding a small regularization parameter $D(P + \delta I, -\delta I, -\delta I)$ (in Eq. (16)) as default $\delta \approx 10^{-6}$. We could then solve it with the aforementioned ADMM procedure to obtain a candidate solution, denoted as $\hat{t}$ and recover the exact solution $t$ from the perturbed KKT conditions $(K + \Delta K)\hat{t} = g$ by iteratively solving:

$$(K + \Delta K)\Delta \hat{t}^k = g - K \hat{t}^k. \tag{17}$$

where $\hat{t}^{k+1} = \hat{t}^k + \Delta \hat{t}^k$ and it converges to $t$ very quickly in practice Stellato et al. (2020) for only one backward- and one forward-solve. Thus our BPQP method solves backward pass problems in a general but efficient way.

## 4.3 EXAMPLE: DIFFERENTIABLE QP AND SOCP

Below we provide examples for differentiable QP and SOCP oracles (i.e. solutions) using BPQP. The general procedure is to first write down KKT matrix of the original decision making problem. And then apply Theorem 1. Assuming the optimal solution $z^\star$ is already obtained in forward pass.

**Differentiable QP** With a slight abuse of notation, given the standard QP problem with parameters $P, q, A, b, G, c$ as in Eq. (10). The result is exactly the same as OptNet Amos & Kolter (2017) since both approaches are for accurate gradients. But BPQP is capable of efficiently solving large-scale QP forward-backward pass via ADMM Stellato et al. (2020), as shown in Section 5.1.

$$\begin{aligned} \nabla_Q \mathcal{L} &= \tfrac{1}{2} \left( \tilde{z} z^{\star T} + z^\star \tilde{z}^T \right) & \nabla_q \mathcal{L} &= \tilde{z} & \nabla_A \mathcal{L} &= \tilde{\nu} z^{\star T} + \nu^\star \tilde{z}^T \\ \nabla_b \mathcal{L} &= -\tilde{\nu} & \nabla_{G_+} \mathcal{L} &= D(\lambda_+^\star) \tilde{\lambda} z^{\star T} + \lambda_+^\star \tilde{z}^T & \nabla_{c_+} \mathcal{L} &= -D(\lambda_+^\star) \tilde{\lambda} \end{aligned} \tag{18}$$

And $[\tilde{z}, \tilde{\nu}, \tilde{\lambda}]$ solves

$$\underset{\tilde{z}}{\text{minimize}} \; \frac{1}{2} \tilde{z}^\top P \tilde{z} + \frac{\partial \mathcal{L}}{\partial z^\star}^\top \tilde{z} \quad \text{s.t. } A\tilde{z} = 0, \; G_+ \tilde{z} = 0. \tag{19}$$

---

[1] $G_+ = G(z_+^\star)$ has the same row of active-set as $g(z_+^\star) = 0$, $z \in \mathbb{R}^{m_+}$. $m_+$ is the number of active sets.

**Differentiable SOCP** The second-order cone programming (SOCP) of our interest is the problem of robust linear program Bennett & Mangasarian (1992):

$$\underset{z}{\text{minimize}} \; q^\top z \quad \text{s.t.} \; a_i^\top z + \|z\|_2 \le b_i \; i = 1, 2, ..., m. \tag{20}$$

where $q \in \mathbb{R}^d$, $a_i \in \mathbb{R}^d$, and $b_i \in \mathbb{R}$. With $m$ inequality constraints in $L2$ norm, we give the gradients w.r.t. above parameters.

$$\nabla_q \mathcal{L} = \tilde{z} \quad \nabla_{a_{i+}} \mathcal{L} = \lambda_{i+}^\star \tilde{z} + \lambda_{i+}^\star \tilde{\lambda}_i z^\star \quad \nabla_{c_{i+}} \mathcal{L} = \tilde{\lambda}_i, \; i = 1, 2, ..., m. \tag{21}$$

And $[\tilde{z}, \tilde{\nu}, \tilde{\lambda}]$ are given by ($t_1 = \sum_i \lambda_{i+}^\star$ and $t_0 = \|z^\star\|_2$)

$$\underset{\tilde{z}}{\text{minimize}} \; \frac{1}{2}\tilde{z}^\top \left( \frac{t_1}{t_0}\mathbb{I} - \frac{t_1}{t_0^3}z^\star z^{\star\top} \right) \tilde{z} + \frac{\partial \mathcal{L}}{\partial z^\star}\tilde{z} \quad \text{s.t.} \; (a_{i+}^\top + \frac{1}{t_0}z^\star)^T \tilde{z} = 0, \; i = 1, 2, ..., m. \tag{22}$$

## 5 EXPERIMENTS

In this section, we present several experimental results that highlight the capabilities of the BPQP. To be precise, we evaluate for (i) large-scale computational efficiency over existing solvers on random-generated constrained optimization problems including QP, LP, and SOCP, and (ii) performance on real-world end-to-end portfolio optimization task that is challenging for existing predict-then-optimize approaches.

### 5.1 SIMULATED LARGE-SCALE CONSTRAINED OPTIMIZATION

We randomly generate three datasets (e.g. simulated constrained optimization) for QPs, LPs, and SOCPs respectively. The datasets cover diverse scales of problems. The problem scale includes $10 \times 5$, $50 \times 10$, $100 \times 20$, $500 \times 100$ (e.g., $10 \times 5$ represents the scale of 10 variables, 5 equality constraints, and 5 inequality constraints). Please refer to more experiment details in Appendix A.4.

**QPs Dataset** The format of generated QPs follows Eq. (12) to which the notations in the following descriptions align. We take $q$ as the learnable parameter to be differentiated and $\mathcal{L} = \mathbf{1}^\top z^\star$ in Eq. (9). To generate a positive semi-definite matrix $P$, $P'^\top P' + \delta I$ is assigned to $P$ where $P' \in \mathbb{R}^{d \times d}$ is a randomly generated dense matrix, $\delta I$ is a small regularization matrix, and $\delta = 10^{-6}$. Potentially, we set $c = Gz'$, $G \in \mathbb{R}^{m \times n}$, $z' \in \mathbb{R}^n$ to avoid large slackness values that lead to inaccurate results. All other random variables are drawn i.i.d. from standard normal distribution $N(0, 1)$.

**LPs Dataset** The LP problems are generated in the format below

$$\underset{z}{\text{minimize}} \; \theta^T z + \epsilon\|z\|_2^2 \quad \text{s.t.} \; Az = b, Gz \le h. \tag{23}$$

where $\theta \in \mathbb{R}^d$ is the learnable parameter to be differentiated, $z \in \mathbb{R}^d$, $A \in \mathbb{R}^{n \times d}$, $b \in \mathbb{R}^n$, $G \in \mathbb{R}^{m \times d}$, $h \in \mathbb{R}^m$ and $\epsilon \in \mathbb{R}_+$. All random variables are drawn from the same distribution as the QPs dataset. It is noteworthy that it contains an extra item $\epsilon\|z\|_2^2$ compared with traditional LP. This item is added to make the optimal solution $z^\star$ differentiable with respect to $\theta$. Without this item, $P(z^\star, \nu^\star, \lambda^\star)$ is always zero and thus the left-hand side matrix becomes singular in Eq. (8). This is a trick adopted by previous work Wilder et al. (2019). CVXPY will reformulate the problem to a cone program and can handle this issue internally. So $\epsilon$ is set to 0 for CVXPY. For other differentiable optimizers, $\epsilon = 10^{-6}$ as default.

**SOCPs Dataset** For SOCP in Eq. (20), we consider a specific simple case, i.e. $a_i = 0 \; \forall i$ and this relaxations results in $m = 1$. As in QP and LP, we take $q$ as differentiable parameter and set loss function $\mathcal{L} = \mathbf{1}^\top z^\star$, but all variables are drawn i.i.d. from standard Gaussian distribution $N(0, 1)$.

**Compared Methods** To demonstrate the effectiveness of BPQP, we evaluate the efficiency and accuracy of state-of-the-art differentiable convex optimizers, as well as **BPQP**, on the datasets mentioned above. The following methods are compared: **CVXPY** Agrawal et al. (2019b), **qpth/OptNet** Amos & Kolter (2017), **Alt-Diff** Sun et al. (2022), **JAXOpt** Blondel et al. (2021) and **Exact** Gould et al. (2021). Exact adopts the same algorithm as BPQP for the forward pass, but attempts to calculate exact gradients using direct matrix inversion on the KKT matrix during the backward pass.

**Evaluation and Metrics** To evaluate the efficiency of the compared methods, the runtime in seconds is used for each forward pass, backward pass, and total process. To evaluate the accuracy, we first get a target solution $z^{\text{Exact}}$ with a high-accuracy method and then calculate the cos similarity with compared methods ($\text{CosSim} = z^{\text{Exact}} \cdot z^{\text{method}_i} / (\|z^{\text{Exact}}\| * \|z^{\text{method}_i}\|)$). We ran each instance 200 times for average and standard deviation (marked in brackets) of the metrics.

Table 1: Efficiency evaluation of methods by runtime in seconds based on 200 runs, with lower numbers indicating better performance.

| dataset | metric | stage size method | Backward | | | | Total(Forward + Backward) | | | |
|---|---|---|---|---|---|---|---|---|---|---|
| | | | 10×5 | 50×10 | 100×20 | 500×100 | 10×5 | 50×10 | 100×20 | 500×100 |
| QP (small) | abs. time (scale 1.0e-04) | Exact | 40.6(±60.3) | 325.4(±280.7) | 3388.8(±540.6) | 37279.4(±2503.0) | 41.1(±60.5) | 333.7(±283.7) | 3440.1 (±554.6) | 37349.3 (±2672.1) |
| | | CVXPY | 39.5(±19.1) | 75.2(±17.1) | - | - | 472.6(±143.2) | 38796.1(±1430.3) | - | - |
| | | qpth/OptNet | 33.3(±9.8) | 35.9(±8.9) | 38.3(±12.3) | - | 851.6(±499.4) | 952.8(±201.0) | 1308.2(±238.0) | - |
| | | Alt-Diff | - | - | - | - | 325.5(±686.4) | 834.7(±475.1) | 4516.3(±1863.2) | 34775.7(±12835.3) |
| | | BPQP | **0.5(±0.1)** | **2.6(±0.5)** | **10.5(±8.4)** | **116.2(±20.4)** | **1.6(±0.6)** | **10.9(±5.4)** | **61.8(±35.3)** | **1632.9(±223.7)** |
| | (scale 1.0e+01) | JAXOpt | 0.5(±0.2) | 0.8(±0.3) | 1.8(±0.8) | 38.3(±8.6) | 0.6(±0.3) | 1.2(±0.5) | 3.2(±1.3) | 72.9(±8.8) |
| LP | abs. time (scale 1.0e-03) | Exact | 1.2(±2.0) | 19.5(±12.4) | 240.5(±36.4) | 2955.6(±131.7) | 1.3(±2.1) | 19.9 (±13.4) | 242.7(±36.4) | 3025.5 (±141.2) |
| | | CVXPY | 4.3(±2.8) | 3.7(±1.0) | 6.1(±2.2) | 25.9(±2.9) | 28.3(±9.0) | 26.1(±6.4) | 45.3(±13.6) | 302.1(±20.7) |
| | | qpth/OptNet | 3.9(±1.1) | 3.7(±1.1) | 4.0(±1.0) | 5.9(±0.9) | 112.2(±24.3) | 106.3(±25.6) | 116.1(±23.0) | 248.5(±58.5) |
| | | BPQP | **0.1(±0.8)** | **0.1(±0.0)** | **0.6(±1.3)** | **4.8(±0.8)** | **0.2(±0.9)** | **0.5(±0.2)** | **2.8(±1.5)** | **74.7(±21.4)** |
| SOCP | abs. time (scale 1.0e-04) | Exact | 2.3(±5.1) | 4.2(±6.3) | 12.6(±23.2) | 110.7(±116.8) | 47.5 (±10.0) | 52.0(±8.4) | 73.3(±24.2) | 300.2 (±119.5) |
| | | CVXPY | 8.8(±0.9) | 8.9(±0.3) | 9.0(±0.6) | **11.1(±0.3)** | 64.1(±5.1) | 80.1(±3.4) | 105.0(±2.9) | 334.3(±3.2) |
| | | BPQP | **0.2(±0.0)** | **0.7(±0.0)** | **2.3(±0.0)** | 53.4(±0.3) | **45.4(±4.9)** | **48.5(±2.6)** | **63.1(±1.6)** | **242.9(±2.7)** |

Table 2: Large-scale comparison of efficiency evaluation of methods by runtime in seconds based on 10 runs, with lower numbers indicating better performance.

| dataset | metric | stage size method | Backward | | | | Total(Forward + Backward) | | | |
|---|---|---|---|---|---|---|---|---|---|---|
| | | | 500×200 | 1500×500 | 3000×1000 | 5000×2000 | 500×200 | 1500×500 | 3000×1000 | 5000×2000 |
| QP (large) | abs. time (scale 1.0e-01) | Exact | 43.6(±3.6) | 78.6(±8.6) | 112.6(±10.0) | 201.5(±15.3) | 44.1(±3.7) | 89.4 (±10.1) | 184.8 (±15.8) | 482.6 (±35.9) |
| | | Alt-Diff | - | - | - | - | 73.6(±19.0) | 197.5(±36.5) | 630.0(±77.8) | 3490.3(±408.4) |
| | | BPQP | **0.2(±0.0)** | **1.7(±0.3)** | **7.4(±0.5)** | **23.7(±1.6)** | **0.7(±0.1)** | **12.5(±1.8)** | **79.6(±6.3)** | **304.8(±22.0)** |

**Results** The results for efficiency evaluation are shown in Table 1. The evaluation covers three typical optimization problems with different problem scales. The results start from the QP dataset. Compared with state-of-the-art accurate methods, BPQP achieves tens to thousands of times of speedup in total time. When the problem becomes large, such as 5000×2000, previous methods fail to generate results. CVXPY is extremely much slower because it reformulates the QP as a conic program and the reformulation is slow and has to be done repeatedly when the problem parameters change Stellato et al. (2020). It is worth noting that BPQP is faster even in the backward pass, where CVXPY and qpth/OptNet share information from the forward pass to reduce computational costs. Sharing this information will limit the available forward solvers and result in a coupled design. Exact falls back to a simpler implementation that does not involve sharing information between designs. It solves the KKT matrix (i.e., Eq. (9)) in the backward pass via a matrix inverse method without relying on information from the forward pass. Although Exact uses a relatively efficient implementation in the forward pass (i.e., a first-order method, same as BPQP), the fallback backward implementation becomes a bottleneck for efficiency. The results of the LP dataset lead to similar conclusions as those of the QP dataset.

In the evaluation of the SOCP dataset, qpth/OptNet and Alt-Diff focus on QP and are excluded from this non-QP setting. Due to the specialty of SOCP, CVXPY does not require problem reformulation into conic programs, giving it an advantage. BPQP still outperforms other options in terms of total time across all problem scales.

Table 3: Backward accuracy of methods on simulated QP and non-QP(SOCP) dataset

| method | QP | | | | | SOCP | |
|---|---|---|---|---|---|---|---|
| | BPQP | CVXPY | qpth/OptNet | Alt-Diff | JAXOpt | BPQP | CVXPY |
| Avg. CosSim. | **0.992(±0.092)** | 0.520(±0.48) | 0.989(±0.12) | 0.985(±0.11) | 0.831(±0.14) | **1.00(±1.8e-013)** | 1.00(±1.3e-012) |

The accuracy evaluation results are shown in Table 1. In the forward pass, all solvers give nearly the same results, which are not shown in the table. When evaluating the backward accuracy, we use a matrix inverse method with high precision to solve Eq. (9) directly to get a target solution(i.e. $z^{Exact}$) and compare solutions from evaluated methods against it. The $CosSim.$ is relatively higher

than that in the forward pass due to accumulated computational errors. Among them, the $CosSim.$ of our method BPQP is the highest in QP. The $CosSim.$ of all methods are small enough for SOCP.

## 5.2 Real-world End-to-End Portfolio Optimization

Portfolio optimization is a fundamental problem for asset allocation in finance. It involves constructing and balancing the investment portfolio periodically to maximize profit and minimize risk. We now show how to apply BPQP to the problem of end-to-end portfolio optimization(more experiment details in Appendix A.5).

**Mean-Variance Optimization (MVO)** Markowitz (1952) is a basic portfolio optimization model that maximizes risk-adjusted returns and requires long only and budget constraints.

$$\underset{w}{\text{maximize}} \; \mu^\top w - \frac{\gamma}{2} w^\top \Sigma w \; \text{ subject to } \mathbf{1}^\top w = 1, \; w \geq 0. \tag{24}$$

where variables $w \in \mathbb{R}^d$ represent the portfolio weight, $\gamma \in \mathbb{R} > 0$, the risk aversion coefficient, and $\mu \in \mathbb{R}^d$ the expected returns to be predicted. We built an ML predictor to approximate expected returns. The covariance matrix, $\Sigma$, of all assets can be learned end-to-end by BPQP. However, it preserves a more stable characteristic than returns in time-series Lux & Marchesi (2000). Therefore, we set it as a constant.

**Benchmarks** We evaluate BPQP based on the most widely used predictive baseline neural network, MLP. For the learning approach, we compared the separately two-stage(**Two-Stage**) and end-to-end learning approaches(**qpth/OptNet**). The optimization problem in the experiment has a variable scale of 500, which cannot be handled by other layers based on CVXPY and JAXOpt. We found the tolerance level for truncation in Alt-Diff hard to satisfy the 500 inequality constraints and yield a relatively longer training time (588 minutes per training epoch) than the above benchmarks. Our implementation substantially lowers the barrier to using convex optimization layers.

Table 4: Prediction and decision(portfolio) metrics evaluation of different methods in portfolio optimization. Speed is evaluated by training time per epoch (minute).

|  | Prediction Metrics | | Portfolio Metrics | | Optimization Metrics | |
| --- | --- | --- | --- | --- | --- | --- |
|  | IC ↑ | ICIR ↑ | Ann.Ret.(%) ↑ | Sharpe ↑ | Regret↓ | Speed↓ |
| Two-Stage | **0.033(±0.004)** | **0.32(±0.03)** | 9.28(±3.46) | 0.65(±0.25) | 0.0283(±0.0271) | **0.11** |
| qpth/OptNet | 0.026(±0.003) | 0.38(±0.12) | 16.54(±7.51) | 1.25(±0.42) | 0.0176(±0.0049) | 21.2 |
| BPQP | 0.026(±0.002) | 0.28(±0.03) | **17.67(±6.11)** | **1.28(±0.43)** | **0.0129(±0.0020)** | 7.7 |

**Results** The overall results are shown in Table 4. As we can see in the prediction metrics, Two-Stage performs best. Instead of minimizing multiple objectives without a non-competing guarantee, Two-Stage only focuses on minimizing the prediction error and thus avoids the trade-off between different objectives. However, achieving the best prediction performance does not equal the best decision performance. BPQP outperforms Two-Stage in all portfolio metrics, although its prediction performance is slightly compromised. qpth/OptNet shows comparable performance with BPQP. But the average training time of BPQP is 2.75x faster than OptNet. These experiments demonstrate the superiority of end-to-end learning, which minimizes the ultimate decision error, over separate two-stage learning.

## 6 Conclusion

We have introduced a differentiable convex optimization framework for efficient end-to-end learning. Based on whether an explicit computational graph is constructed, previous work on differentiable convex optimization layers methods can be categorized into explicit and implicit methods. Explicit methods unroll the iterations of the optimization process, incurring additional costs. Implicit methods can't achieve overall efficiency on both computing the optimal decision variable during the forward pass and solving the KKT matrix during the backward pass. Our work, BPQP, is based on implicit methods. We simplify the backward pass by reformulating it into a simpler decoupled QP problem, which greatly reduces the computational cost in both the forward and backward passes. Extensive experiments on both simulated and real-world datasets have been conducted, demonstrating a considerable improvement in terms of efficiency.

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

# A APPENDIX

## A.1 MAP PREDICT-THEN-OPTIMIZE LOSS

On selecting $\beta$ for portfolio optimization in section 5.2, mathematically $\beta = \frac{\sigma_r^2}{\sigma_y^2} \in (0,1)$ denotes the ratio of variances between the random parameter $y$ and the regret. Since the empirical regret suffers a more severe fluctuation over $y$ ($\sigma_r \gg \sigma_y > 0$) in convex optimization Mandi et al. (2020), prediction error should dominate in the end-to-end loss. However, we use an empiric distribution to approximate the Dirac distribution and set $\beta$ to a small ($\beta = 0.1$ in portfolio optimization experiment) but not zero value.

Under normality assumption

$$\arg\max(p(\theta \mid \text{regret}, y, x)) \propto$$

$$\arg\max \prod_i \frac{1}{\sigma_r \sqrt{2\pi}} e^{-\frac{\text{regret}_i^2}{2\sigma_r^2}} \times \prod_j \frac{1}{\sigma_y \sqrt{2\pi}} e^{-\frac{(y_j - \hat{y}_j)_j^2}{2\sigma_y^2}}, \tag{25}$$

That is

$$\arg\min \sum_i regret_i^2 + \frac{\sigma_r^2}{\sigma_y^2} \sum_j (y_j - \hat{y}_j)^2. \tag{26}$$

## A.2 DIFFERENTIATE THROUGH KKT CONDITIONS USING THE IMPLICIT FUNCTION THEOREM

In this section, we give a detailed discussion on Eq. (9). The sufficient and necessary conditions for optimality for Eq. (1) are KKT conditions:

$$\begin{aligned}
\nabla f(z^\star) + \nabla h(z^\star)\nu^\star + \nabla g(z^\star)\lambda^\star &= 0 \\
h(z^\star) &= 0 \\
D(\lambda^\star)(g(z^\star)) &= 0 \\
\lambda^\star &\geq 0,
\end{aligned} \tag{27}$$

Applying the Implicit Function Theorem to the KKT conditions and let $P(z^\star, \nu^\star, \lambda^\star) = \nabla^2 f(z^\star) + \nabla^2 h(z^\star)\nu^\star + \nabla^2 g(z^\star)\lambda^\star$, $A(z^\star) = \nabla h(z^\star)$ and $G(z^\star) = \nabla g(z^\star)$ yields to Eq. (8). We can then backpropagate losses by solving the linear system. In practice, however, explicitly computing the actual Jacobian matrices $\frac{\partial z^\star}{\partial y}$ is not desirable due to space complexity; instead, we product some previous pass gradient vectors $\frac{\partial \mathcal{L}}{\partial z^\star} \in \mathbb{R}^d$, to reform it by noting that

$$\nabla_y \mathcal{L} = \begin{bmatrix} \frac{\partial z^\star}{\partial y}, & \frac{\partial \lambda^\star}{\partial y}, & \frac{\partial \nu^\star}{\partial y} \end{bmatrix} \begin{bmatrix} \left(\frac{\partial \mathcal{L}}{\partial z^*}\right)^\top \\ 0 \\ 0 \end{bmatrix}, \tag{28}$$

The first term of left hand side is the transposed solution of Eq. (8) and above can be reformulated as

$$\nabla_y \mathcal{L} = [q,\ c,\ b] \underbrace{\begin{bmatrix} P(z^\star, \nu^\star, \lambda^\star) & D(\lambda^\star)G(z^\star) & A(z^\star) \\ G(z^\star)^\top & D(g(x^\star)) & 0 \\ A(z^\star)^\top & 0 & 0 \end{bmatrix}^{-1} \begin{bmatrix} -\left(\frac{\partial \mathcal{L}}{\partial z^*}\right)^\top \\ 0 \\ 0 \end{bmatrix}}_{\text{BPQP solution: } [\tilde{z}, \tilde{\lambda}, \tilde{\nu}]^\top}. \tag{29}$$

## A.3 PRESERVE KKT NORM GRADIENTS

In a typical optimization algorithm, each stage of the iteration gives primal-dual conditions $r^{(k)}$, we follow the procedures of BPQP and solve the corresponding QP problem $\mathcal{Q}^{(k)}$ to define general gradients $\nabla_y \mathcal{L}^{(k)}$. The key difference here is that instead of using the optimal solution to derive BPQP, we plug in the intermediate points. By IFT,

$$dr^{(k)} = K^{(k)}[dz, d\lambda, d\nu]^\top + \frac{\partial r^{(k)}}{\partial y} dy = 0. \tag{30}$$

where $K^{(k)}$ is the Hessian matrix (KKT matrix) at points $(z_k, \lambda_k, \nu_k)$. The general gradients $\nabla_y \mathcal{L}^{(k)}$ is given by $dr^{(k)} = 0$ and therefore $\|r^{(k)}\| = C_k$ preserves KKT norm.

## A.4 SIMULATION EXPERIMENT

### COMPARED METHODS

In Section 5.1, we randomly generate simulated constrained optimization datasets with uniform distributions and varying scales. We use these datasets to evaluate the efficiency and accuracy of state-of-the-art differentiable convex optimizers as well as BPQP. The methods of comparison briefly introduced previously are now detailed below:

**CVXPY** is a universal differentiable convex solver Diamond & Boyd (2016); Agrawal et al. (2019b;a). SCS O'Donoghue et al. (2016); O'Donoghue (2021) solver is employed to accelerate the gradients calculation process.

**qpth/OptNet** qpth is a GPU-based differentiable optimizer, OptNet Amos & Kolter (2017) is a differentiable neural network layer that wraps qpth as the internal optimizer.

**BPQP** is our proposed method. Its forward and backward passes are implemented in a decoupled way. It adopts the OSQP Stellato et al. (2020) as the forward pass solver. In the backward pass, it reformulates the backward pass as an equivalent simplified equality-constrained QP. OSQP is also adopted in the backward pass to solve the QP.

**Exact** uses the same forward pass solver as BPQP. The optimization algorithm used for the forward pass is the OSQP Stellato et al. (2020), which is a first-order optimization algorithm that does not share differential structure information. In the backward pass, without using reformulation via BPQP, the Eq. (9) are solved using the matrix inversion method like Gould et al. (2021). As a result, this approach fails to achieve overall efficiency.

**JAXOpt** Blondel et al. (2021) is an open-sourced optimization package that supports hardware accelerated, catchable training and differentiable backward pass. Optimization problem solutions can be differentiated with respect to their inputs either implicitly or via autodiff of unrolled algorithm iterations.

**Alt-Diff** Sun et al. (2022) adopts ADMM in specializing in solving QP problems with exact solutions as well as gradients w.r.t. parameters.

### HARDWARE SETTING

All results were obtained on an unloaded 16-core Intel(R) Xeon(R) CPU E5-2630 v3 @ 2.40GHz. qpth runs on an NVIDIA GeForce GTX TITAN X.

### CHOICE OF SOLVERS OF BPQP

BPQP decoupled the forward and backward pass and provides flexibility of choosing solvers. Normally, the first-order solver is greatly preferred when the problem scale becomes large and is also robust for small problem scale. Therefore, the first-order solver is a good enough default value, which is also employed by our framework and experiments.

## A.5 PORTFOLIO OPTIMIZATION EXPERIMENT

### STATISTICAL RISK MODEL (SRM)

is used to generate the covariance matrix of MVO in Section 5.2. It takes the first 10 components with the largest eigenvalues by applying PCA on stock returns in the last 240 trading days. SRM shows the best performance of the traditional data-driven approach for learning latent risk factors.

### DATASET & METRICS

This section provides a more detailed introduction to the datasets and metrics used in the experiments described in Section 5.2. The dataset is from Qlib Yang et al. (2020) and consists of 158 sequences,

each containing OHLC-based time-series technical features Beyaz et al. (2018) from 2008 to 2020 in daily frequency. Our experiment is conducted on CSI 500 universe which contains at most 500 different stocks each day.

For the predictive metrics, we evaluate IC (Information Coefficient) and ICIR (IC Information Ratio) of predictive model baselines. IC measures the correlation coefficient between the predicted stock returns $\hat{y}$ and the ground truth $y$. At each timestamp $t$, $IC^{(t)} = corr(\hat{y}^{(t)}, y^{(t)})$ in which

$$corr(\mathbf{x}, \mathbf{y}) = \frac{\sum_i (x_i - \bar{x})(y_i - \bar{y})}{\sqrt{\sum_i (x_i - \bar{x})^2 \sum_i (y_i - \bar{y})^2}}.$$

We report average IC across instances. $ICIR = \frac{mean(IC)}{std(IC)}$ measures both the average and stability of IC. A well-trained predictive model is expected to have higher IC and ICIR. For portfolio metrics, which measure the performance of investment strategies in the real market, we include two key indicators, $Ann.Ret.$ (Annualized Return) and $Sharpe$ (Sharpe Ratio) , which are the ultimate criteria widely used in quantitative investment. $Ann.Ret.$ indicates the return of given portfolios each year. $Sharpe = \frac{Ann.Ret.}{Ann.Vol.}$ in which $Ann.Vol.$ indicates the annualized volatility. To achieve higher $Sharpe$, portfolios are expected to maximize the total return and minimize the volatility of the daily returns. Transaction costs are not considered in our portfolio metrics to align with the regret loss and more stably demonstrate the effectiveness of end-to-end learning without being distracted by unconsidered random factors.

### COMPARED METHODS

Here is a more detailed explanation of the compared methods in this experiment.

**Two-Stage** separately learns a prediction MLP model to predict expected returns (i.e. $\mu$) and then generates decisions based on Eq. (24). All other methods below share the same prediction MLP model and only differ in the learning paradigm.

**qpth/OptNet** performs similar to BPQP, but with sightly lower performance in portfolio metrics and regret as it approaches exact gradient with a lower accuracy, shown in Table 3.

**BPQP** is our proposed method. All the accurate approaches (e.g. CVXPY, JAXOpt) have similar high-quality solutions in both forward and backward passes and are expected to have similar performance. Among them, only BPQP can efficiently handle the problem size of 500 variables(refer to Table 1), and thus BPQP are selected.

BPQP is trained using the loss function described in Section A.1.

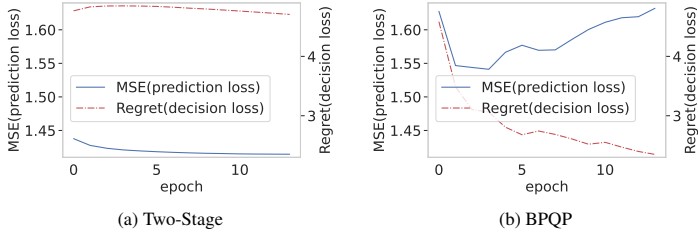

(a) Two-Stage            (b) BPQP

Figure 2: The prediction and decision error/loss of methods with different objectives

To gain a deeper understanding of how end-to-end regret loss works, Figure 2 demonstrates the detailed learning curve of Two-Stage and BPQP. For each subfigure, the x-axis represents the number of epochs during training, and the y-axis represents the training loss of prediction and decision, respectively. Two-Stage aims to minimize the prediction loss, which is ultimately smaller than BPQP. However, the decision loss remains at a high level, resulting in a suboptimal decision. BPQP aims to minimize both prediction loss and decision loss. Both losses decrease initially, and then they start to compete in the later epochs. However, the decision error remains at a much lower value than Two-Stage, resulting in better decisions in the final evaluation.

EXPERIMENT SETTING

Here are the detailed search space for model architecture and hyper- parameters.

We use the same tolerance parameters for simulations experiments: Dual infeasibility tolerance: 1e-04, Primal infeasibility tolerance: 1e-04, Check termination interval: 25, Absolute tolerance: 1e-03, Relative tolerance: 1e-03, ADMM relaxation parameter: 1.6, Maximum number of iterations: 4000.

We use a lower tolerance parameter for real-world portfolio optimization experiments, due to the long-only strategy, we do not want small negative weight in the portfolio: Absolute tolerance: 1e-05, Relative tolerance: 1e-05, Dual infeasibility tolerance: 1e-05, Primal infeasibility tolerance: 1e-05.

MLP predictor: feature size: 153, hidden layer size: 256, number of layersr: 3, dropout rate: 0.Training: number of epoch: 30, learning rate: 1e-4, optimizer: Adam, frequency of rebalancing portfolio: 5 days, risk aversion coefficient: 1, early stopping rounds: 5, the inverse of beta (line 112): 0.1.

DC3: hidden size of solver net: 512, max stock size: 530, corrEps: 1e-4, corrTestMaxSteps: 10, softWeightEqFrac: 0.5, corrMomentum: 0.

## A.6 ADDITIONAL RELATED WORKS.

In this section, we discuss additional related works that are approximate, scenario-specific, or focus on different problems.

### A.6.1 LEARN-TO-OPTIMIZE

Learn-to-optimize has relatively low accuracy, which means it can only support some problems with a high error tolerance. DC3 Donti et al. (2021) and ProjectNet Cristian et al. (2023) are research works in this direction. They leverage the universal approximation ability of neural networks and choose error correction algorithms to modify the output solution into the feasible region. Kong et al. (2022) exploits energy-based model for decision-focused learning.

As a comparison to compute for exact gradients, existing work on *Learn-to-Optimize* trains an approximated solver network via SGD (e.g. DC3 Donti et al. (2021)) or RL policy gradients Joshi et al. (2022); Khalil et al. (2017); Ma et al. (2019); Kool et al. (2018) to solve constrained optimization problems that have a true graphical structure e.g. TSP, VRP, Minimum Vertex Cover, Max-Cut, and their variants. Leveraging the strong representation ability of state-of-the-art graph-based networks, RL obtains final solutions or intermediate results to be polished by searching or optimization algorithms. When optimizing convex and hard constraints in real-world scenarios, the underlying graph is typically fully connected and the accuracy tolerance is lowerUysal et al. (2021). This presents a restriction on the widespread usage of works based on approximated solvers.

### A.6.2 SURROGATE LOSS

Surrogate loss is usually task-specific. LODL Shah et al. (2022) proposes a general surrogate loss but requires expensive computation, and thus it is not general enough. Wang et al. (2020) uses the linear low-dimension representation of original convex problem, which is an approximate surrogate loss and would bring loss of accuracy.

### A.6.3 METHODS THAT FOCUS ON DIFFERENT PROBLEMS.

In this section, we will introduce some works that focus on similar but different research problems. Some works, such as Abbas & Swoboda (2021) and Niepert et al. (2021), focus on discrete optimization, which is not the direction we are focusing on. Verma et al. (2023) focuses on the Restless Multi-Armed decision model.

### A.6.4 ADDITIONAL EXPERIMENTS OF APPROXIMATE METHODS.

In this section, we present additional experiments results for a typical learning-based approximate optimizer, DC3. DC3 are trained using the loss function described in Section A.1. We train the

| size | 10×5 | 50×10 | 100×20 | 500×100 |
|---|---|---|---|---|
| forward + backward time(s) | 1.4e-02 | 1.5e-02 | 1.7e-02 | 1.7e-02 |

Table 5: Efficiency evaluation of the learn-to-optimize method DC3 by runtime in seconds.

solver net (i.e. optimizer) with 500 epochs, 10000 samples, and 10 correction Test Max Steps for each type of QP and LP entries.

The computational cost scales linearly with the problem size and the number of model parameters. So, it is very efficient, especially when the scale is large. When the problem size is small (10×5 or 50×10), BPQP is still tens of times faster than DC3. However, DC3 becomes 5-10 times faster than BPQP when the problem size becomes large (500×100). Table 5 is the more detailed result of DC3 for both QP & LP (their problem size and number of model parameters are the same, so the time is nearly the same).

For large-scale real-world portfolio optimization, approximate methods such as DC3 can be a practical solution in terms of efficiency. The experiment results are shown in Table 6. Although DC3 is computationally efficient and thus applicable to large-scale real-world datasets, it performs poorly in decision metrics. This is due to the inaccurate gradient that deteriorates the learned model based on DC3. Therefore, accuracy is an important feature in end-to-end learning.

Table 6: Prediction and decision(portfolio) metrics evaluation of DC3 in portfolio optimization. Speed is evaluated by training time per epoch (minute).

| | Prediction Metrics | | Portfolio Metrics | | Optimization Metrics |
|---|---|---|---|---|---|
| | IC ↑ | ICIR ↑ | Ann.Ret.(%) ↑ | Sharpe ↑ | Speed↓ |
| DC3 | 0.033(±0.001) | 0.31(±0.01) | -0.40(±0.97) | -0.16(±0.60) | 0.43 |

