# OpenReview forum: "BPQP: A Differentiable Convex Optimization Framework for Efficient End-to-End Learning"
_ICLR.cc/2024/Conference — ICLR 2024 Conference Withdrawn Submission_

### Official Review · Reviewer_kf2G · 2023-11-02

**Soundness:** 1 poor
**Presentation:** 4 excellent
**Contribution:** 3 good
**Rating:** 3
**Confidence:** 4

**Summary:**

This paper studies how to use quadratic programming to compute
the linear system arising when implicitly differentiating
the convex optimization problem in eq (1).
This linear system is in equation (9).
Section 4.1 describes how it can be solved with a QP,
Section 4.2 goes on to show how OSQP solves it, and
Section 4.3 discusses how to differentiate QPs and
SOCPs with the method.

**Strengths:**

1. It is a great insight that QP solvers are designed
   to efficiently solve the KKT system! And that OSQP
   is capable of solving them fast. It's reasonable
   that QP solvers are going to be better than general-purpose
   linear-system solvers for these systems as they
   take the structure into consideration.
2. This method has the potential to become the standard
   method for using differentiable QPs as it is simple to
   set up and use. The right software package
   using this seems like it would be widely adopted.
3. The runtime comparisons in Table 1/2 show a notable
   improvement in comparison to many relevant baselines
   and existing methods, although I have a concern in the
   weaknesses if these were properly executed.

**Weaknesses:**

1. The Alt-Diff results reported in Table 1/2 seem very different
   than the originally-published Alt-Diff results.
   Table 1 of the Alt-Diff paper shows that on a QP with
   10k variables, 5k inequality constraints, and 2k equality
   constraints, their total solve time is 154 seconds.
   However Table 2 of this BPQP paper report that Alt-Diff
   takes 3.4k seconds to solve a QP with 5k variables and 2k
   inequality and equality constraints.
   While this could be coming from a hardware difference,
   I would have found it significantly more convincing to
   compare directly on the settings from AltDiff.
2. Table 3 has the surprising result that BPQP has a higher
   accuracy than the direct solver implemented in qpth/OptNet.
   This is surprising because a direct solver should be perfectly accurate.
   It would be good to include a comment on this point.
   I checked the source code of `Large scale QPs and LPs experiment.ipynb`
   and one potential reason this could be happening is that
   the KKT system is being explicitly inverted for the ground-truth
   derivative with:

    ```python
    exact_bb =-(np.linalg.inv(KKT)@np.hstack([np.ones(ndim),np.zeros(nineq),np.zeros(neq)]))[:ndim]
    ```

    rather than calling `np.linalg.solve` on it.
3. jaxopt's runtime seems very slow in Table 1.
   I checked the source code of `Large scale QPs and LPs experiment.ipynb`
   and one reason could be that JAX's JIT compiler is not being used.
   I believe using the JIT will significantly improve
   jaxopt's runtime (but it probably won't be better than BPQP's runtime)
4. The paper states
   "It is worth noting that BPQP is faster even in the backward pass, where
   CVXPY and qpth/OptNet share information from the forward pass to reduce computational costs."
   I checked the source code of `Large scale QPs and LPs experiment.ipynb`
   and believe this appears to be true because the runtime
   reported for BPQP only includes the time OSQP takes to solve
   the problem, but not the setup time or data transfer time.
   This is unfairly compared to calling `.backward()` on
   the other methods and general-purpose solvers which
   have more overhead than just the solve.
   I would find it a fairer comparison to also include the overhead
   of transferring data and setting up the backward QP solver,
   as it seems like an unavoidable part of using a separate solver
   in the backward pass.
5. The experiment and comparison in section 5.2 is disconnected from
   existing DFL work such as in Wilder et al. (2019) and Shah et al. (2022).
   Both of these papers also propose to use differentiable
   optimization for learning a model with the downstream task
   and use cvxpylayers.
   I would have found it more convincing to take exactly the
   open-sourced experimental setups from one of these papers and
   replace the cvxpylayers call with BPQP and show that it is improved.
6. Even though the paper makes comparisons to the more general conic
   solvers in cvxpylayers,
   the method of using a QP to solve the implicit derivative will
   not work in general to replace the backward pass in cvxpylayers.
   Otherwise it would be a really good idea to add that!
   The implicit derivative of more conic problems,
   which is shown in [this paper](https://arxiv.org/abs/1904.09043),
   is in general not computable with a QP solver.
   It would be useful to have a brief section discussing this limitation.

**Questions:**

Overall I am positive about the methodological contribution of
the paper but have some experimental concerns that I pointed out
in the weaknesses. I am optimistic that most/all of my concerns
are correctable and that doing so will result in a scientifically
valid paper and contribution.
I am extremely open to discussing these points and re-evaluating
my score during the discussion period.

---

### Official Review · Reviewer_itcs · 2023-11-03

**Soundness:** 3 good
**Presentation:** 3 good
**Contribution:** 2 fair
**Rating:** 6
**Confidence:** 4

**Summary:**

This paper deals with decision-focused learning. Existing DFL methods are either restricted to specific problems or suffer from low efficiency in the backward pass. Specifically, to improve the efficiency of the backward pass, this paper reformulates backward pass as a quadratic programming problem. This avoids the inversion of the KKT matrix and uses ADMM for large-scale gradient computation. Experiments are conducted on quadratic programming, linear programming, second-order cone programming, and a real-world portfolio optimization problem. Empirical experiments show that the proposed method can improve computational efficiency significantly.

**Strengths:**

The paper is written well.

Scalability is a critical issue that hinders decision-focused learning in large-scale real-world applications. This paper proposes a reasonable solution to avoid the inversion of the KKT matrix. Empirical results also verify that the proposed method improves the computational efficiency of decision-focused learning significantly.

**Weaknesses:**

The novelty of the method is relatively limited. Theoretically, the time complexity of the proposed method is still at the same level of existing methods.

**Questions:**

Does the implementation support parallel GPU computation?

In the introduction, the paper claims cosplayers (Agrawal et al) are designed for specific problems. However, cvxpylayers can be applied for any convex objective which is in the same scope as the proposed method.

---

### Official Review · Reviewer_EcTW · 2023-11-05

**Soundness:** 3 good
**Presentation:** 2 fair
**Contribution:** 2 fair
**Rating:** 6
**Confidence:** 2

**Summary:**

This paper proposes end to end optimization for two-stage real world problems of predict-then-optimize. The paper claims that predict-then-optimize may not aligned with ultimate goal of optimizing the objective. They propose a methodology a backward propagation for their simplified solution (which is arrived by using KKT, Implicit Function Theorem -- IFT ). They experimented and compared their proposed framework with state-of-art optimization algorithms in synthetic and modern portfolio optimization problem.

**Strengths:**

1. Most of the real problems include a prediction task to later use in an optimization task. I think this paper addresses a significant problem which could be applicable to many domains.
2. To the best of our knowledge, the proposed methodology is quite novel borrowing from KKT, IFT to build a backward propagation step to their end to end optimization problem.

**Weaknesses:**

1. The proposed method should be better explained. There is a lot of terms, prior background information that a reader needs to have to be able follow the proposed methodology. Please include some background information about KKT, IFT and more explanation on how you end up with the equations in the paper.
2. I appreciate the paper running their proposed method in a real-world portfolio optimization problem. However, I believe the prediction task in the modern portfolio is too simplistic -- learning the correlation between different strategies. I think applying this problem on harder prediction task would also add value to the paper.

Small point:
3. Please fix the citations in the paper. They are not correct.

**Questions:**

1. The regret is defined in the paper but not used later on. Is there a relationship between the end to end loss and regret ? Can you also provide a bound on the regret with your method ?

---

### Official Review · Reviewer_yny3 · 2023-11-08

**Soundness:** 1 poor
**Presentation:** 2 fair
**Contribution:** 3 good
**Rating:** 3
**Confidence:** 5

**Summary:**

This paper provides a fast procedure for differentiable convex optimization in neural networks that formulates the backward pass as a QP that is decoupled from the forward pass; this allows the forward pass to be solved using fast-off-the-shelf solvers and the backward pass to be solved using a fast off-the-shelf QP solver (OSQP), decreasing overall runtime. The procedure is demonstrated empirically both in the standalone setting of differentiating through an optimization layer (for LPs, QPs, and SOCPs) and for a predict-then-optimize setting (with the decision-making problem given by a QP), and demonstrates superior runtime compared to prior approaches.

**Strengths:**

The core idea here is simple yet potentially powerful: While past implicit differentiation-based optimization layers have (in practice) aimed to reduce runtime by facilitating information-sharing between the forward and backward passes, an alternative approach is to truly decouple the forward and backward passes and formulate each in a way that is amenable to using fast state-of-the-art off-the-shelf solvers. While the information-sharing approach is theoretically nice and could be done for the off-the-shelf solvers, it does require diving into the details of the mathematical formulations for each new type of solver tried in the forward pass, which may slow adoption of faster solvers or further hinder usability by practitioners (adding to the already copious usability challenges faced by differentiable optimization layers). Providing a way to easily "slot in" off-the-shelf solvers allows us in practice to readily benefit from the existing speed (and potential future improvements thereof) easily and quickly.

The speedups in the experimental results as presented are quite impressive.

**Weaknesses:**

Soundness: The soundness of the submission is unfortunately unclear, both in terms of theory and experimental results.
* Theory: The core theorem, Theorem 1, is presented without proof.
* Theory: There is lack of clarification as to why the formulation presented in Theorem 1 is faster to solve than the backward pass KKT system presented in OptNet (for the case where the forward pass is a QP).
* Theory: Since the proposed framework is presented as applicable to convex optimization problems in general (not just QPs in the forward pass), it is necessary to show why the backward pass of those problems can also be formulated as a QP, which is not done.
* Theory: Theorem 1 - how can it be enforced that $c+ = 0, b = 0$? Is there (e.g.) a rearrangement of the input variables to include constant terms, and if so, how is that incorporated into the gradient formulations?
* Experiments: In the first set of experiments (Section 5.1), the comparison between BPQP and Exact is unfair - the fair comparison rather than having Exact solve the KKT matrix via matrix inverse would be to differentiate through the fixed point of ADMM equations (which could benefit from information sharing). Without this fair comparison, it is not possible to evaluate the strength of the submission - is the benefit really coming from the reformulated backward pass, or is it just coming from leveraging OSQP in the forward pass rather than the research code presented in e.g. OptNet?
* Experiments: The LP results headlined in the introduction (21.17× faster) are misleading, since the problems passed to the different solvers are not the same (one version includes a quadratic term, and the other does not). For the fairest comparison in this case, either the same LP formulation should be used as input to all solvers, or the cost of the reformulation process within CVXPY should be removed in the timing comparisons (since adding the QP term for the other solvers is somewhat of an "external reformulation" that is not being accounted for).
* Experiments: Is it unclear where the discrepancy in the headline speedup result for standalone QPs (16.17× speedup) vs. the speedup demonstrated in the end-to-end portfolio optimization experiment (2.75×) comes from.

Strength of results: In principle, the most exciting part of this submission is the ability to reformulate *general* differentiable convex optimization problems within this framework to enable faster performance. However, in addition to potential soundness issues in theory (see above), the performance on these more general settings is actually not particularly impressive - notably, the headline result for speedup in the SOCP setting is only 1.67×, which is not nothing, but is at the level that could be explained by differences in hardware, imprecise timing comparisons, fully optimized implementations vs. research implementations of the same code, etc.

Imprecision/inaccuracies in summary of prior literature and subsequent formulations: The introduction, related work, and background conflate two conceptually similar but different approaches to bridging prediction and optimization (both in terms of discussion and the way the end-to-end model learning problem is formulated). The predict-then-optimize approach (Elmachtoub & Grigas 2020) and subsequent works use regret minimization as the objective, comparing against the objective value of the decision implied by the prediction against the objective value under ground truth values. The task-based end-to-end model learning approach (Donti, Amos, Kolter 2017) and subsequent works use an objective that computes the cost of the decision implied by the prediction, when that decision is enacted on the ground truth (this is the line of approaches that tend to use differentiable optimization). The introduction conflates between these two lines of work (notably, many direct follow-ones from Elmachtoub & Grigas 2020 do *not* use differentiable optimization, despite the implication in the introduction), and the background section mixes between the two types of formulation (which would be fine if the goal of this paper were to introduce a new formulation rather than building directly off the previous ones, but that is not the stated goal).

Minor points on writing:
* Generally, it is unclear to me why the submission is framed up front in terms of predict-then-optimize when the goal is to provide a faster differentiable optimization layer. Predict-then-optimize is simply one use case of having a differentiable optimization layer. The framing of the paper could be improved to be more consistent with its actual contributions.
* Typos and typesetting issues should be fixed (e.g., please use \citep{} for citations).

**Questions:**

* Please clarify or provide additional information to resolve the Soundness issues presented in "Weaknesses" above.
* Why is the performance improvement on SOCPs not greater than demonstrated, given that the margin for improvement is in principle much greater than for LPs and QPs?

**Details Of Ethics Concerns:**

I believe this submission may suffer from minor plagiarism of text, in a way that does not necessarily affect my assessment of the quality of the core ideas of the submission, but which I wanted to flag nonetheless.

In particular, I believe the submission plagiarizes some wording and notation from the following paper, despite this paper not actually being cited. (Strangely, other papers from the same authors are cited despite their much more tenuous connection in some cases.)
> Donti, Priya, Brandon Amos, and J. Zico Kolter. "Task-based end-to-end model learning in stochastic optimization." Advances in neural information processing systems 30 (2017).

Comparison 1:
* BPQP Section 3: "Specifically, let $(x \in \mathcal{X}, y \in \mathcal{Y}) \sim \mathcal{D}$ denote standard input-output pairs drawn from the real and unknown distribution $\mathcal{D}$."
* Donti et al. (2017), Section 3: "Let $(x \in \mathcal{X}, y \in \mathcal{Y}) \sim \mathcal{D}$ denote standard input-output pairs drawn from some (real, unknown) distribution $\mathcal{D}$."

Comparison 2:
* BPQP adopts $z^\star_{\hat{y}}$ as the definition for the optimal decision assuming the prediction is ground truth (Equation 1).
* Donti et al. (2017) presents similar notation ($z^\star(x; \theta)$ in Equation 2) to mean the same thing. This is not, e.g., shared with the (Elmachtoub & Grigas 2020) line of work.

Comparison 3: See the formulation of Equation (6) in BPQP vs. the formulation of Equation (6) in Donti et al. (2017), which present the chain rule required when differentiating through the argmin operator. While BPQP differentiates with respect to $y$ and Donti et al. (2017) differentiates through $\theta$, the equations and their display are very similar, despite this display not necessarily being common or propagated to follow-ons of Donti et al. (2017).

I present these examples not because they are particularly egregious in and of themselves, but because they striking and subtle (I only caught them due to my familiarity with Donti et al. (2017)). I am not able to assess whether there are other instances of similarly subtle (or maybe egregious) plagiarism exhibited in this work. If the plagiarism extends to just the three instances above, then it is potentially borderline - but if it is indicative of a broader transgression with respect to other papers as well, then that is worrying.